# On the identification of differentially-active transcription factors from ATAC-seq data

**Felix Ezequiel Gerbaldo**[1☯]**, Emanuel Sonder**[1,2,3,4☯]**, Vincent Fischer**[5]**, Selina Frei**[5]**, Jiayi Wang**[3]**, Katharina Gapp**[5]**, Mark D. Robinson**[3,4]**, Pierre-Luc Germain**[1,3,4]*

**1** Computational Neurogenomics, D-HEST Institute for Neurosciences, Zürich, Switzerland, **2** Systems Neuroscience, D-HEST Institute for Neurosciences, Zürich, Switzerland, **3** Department of Molecular Life Sciences, University of Zürich, Zürich, Switzerland, **4** SIB Swiss Institute of Bioinformatics, University of Zurich, Switzerland, **5** Epigenetics and Neuroendocrinology, D-HEST Institute for Neurosciences, Zürich, Switzerland

☯ These authors contributed equally to this work.
* pierre-luc.germain@hest.ethz.ch

## Abstract

ATAC-seq has emerged as a rich epigenome profiling technique, and is commonly used to identify Transcription Factors (TFs) underlying given phenomena. A number of methods can be used to identify differentially-active TFs through the accessibility of their DNA-binding motif, however little is known on the best approaches for doing so. Here we benchmark several such methods using a combination of curated datasets with various forms of short-term perturbations on known TFs, as well as semi-simulations. We include both methods specifically designed for this type of data as well as some that can be repurposed for it. We also investigate variations to these methods, and identify three particularly promising approaches (a chromVAR-limma workflow with critical adjustments, monaLisa and a combination of GC smooth quantile normalization and multivariate modeling). We further investigate the specific use of nucleosome-free fragments, the combination of top methods, and the impact of technical variation. Finally, we illustrate the use of the top methods on a novel dataset to characterize the impact on DNA accessibility of TRAnscription Factor TArgeting Chimeras (TRAFTAC), which can deplete TFs—in our case NFkB—at the protein level.

## Author summary

Transcription factors regulate gene expression by binding sites in the genome that often harbor a specific DNA motif. The collective accessibility of these motif-matching regions, measured by technologies such as ATAC-seq, can be used to infer the activity of the corresponding transcription factors. Here we use curated datasets of 11 TF-specific perturbations as well as 116 semi-simulated datasets to benchmark various methods for identifying factors that differ in activity between experimental conditions. We investigate important variations in the analysis and make recommendations pertaining to such analysis. Finally,

**Data Availability Statement:** All code underlying the analyses and figures are made available in the github repository https://github.com/ETHZ-INS/

DTFAB. In addition, new raw sequencing data has being uploaded to the GEO repository with accession ID GSE260504, and semi-simulated data has been deposited on Zenodo under DOIs 10.5281/zenodo.10732704, 10.5281/zenodo.10781849, 10.5281/zenodo.10781759, and 10.5281/zenodo.10781109.

**Funding:** This work was supported by research grants (ETH-25 02-2 to PLG and 23-2 ETH-015 to KG) from the Swiss Federal Institute of Technology (ETH Zurich). The salary of ES is paid by the ETH-25 02-2 grant. The Gapp lab received funding from a SNF PR00P3_201543 and the Swiss State Secretariat for Education, Research and Innovation (SERI) under contract number MB22.00037. The funders had no role in study design, data collection and analysis, decision to publish, or preparation of the manuscript.

**Competing interests:** The authors have declared that no competing interests exist.

we illustrate the application of the top methods to characterize the effects of a novel method for perturbing transcription factors at the protein level.

## 1 Introduction

Transcription factors (TFs) regulate transcription by binding specific sites in the genome, which often harbor a corresponding DNA motif. Identifying which TFs regulate a given transcriptional signature is critical to a mechanistic understanding of gene expression. The expression of the TFs themselves can sometimes be a useful indicator of their involvement, especially between cell types and developmental stages. However, it is often a poor proxy for their activity in dynamic settings, where their activity is often regulated post-transcriptionally. For example, the TF ATF4 is critical to the integrated stress response, and its mRNA is already present in cells but its translation is blocked by an upstream uORF until a translation initiation factor is modified [1, 2]. Another example is the cAMP-response element binding protein (CREB), which is fundamental to a number of processes: CREB is already translated but its phosphorylation strongly enhances its DNA binding ability and allows for the recruitment of co-factors [3]. Further common examples are nuclear receptors, which in general are outside the nucleus until activated by their ligand, upon which they translocate to the nucleus to affect transcription [4]. It is therefore critical to have means of measuring TF activity independently of their expression.

An obvious approach is to profile TF binding through ChIP-seq or derivative technologies, which is however time-consuming, can only be done one TF at a time, and depends on the availability of high-quality antibodies. DNA accessibility instead promises a less biased approach for the discovery of TFs underlying a change in cellular phenotype. In particular, the Assay for Transposase-Accessible Chromatin with sequencing (ATAC-seq) has emerged as a simple, fast accessibility assay that requires little starting material [5, 6], being even amenable to single-cell analysis [7]. Since TF binding is associated with DNA accessibility, such assays offer genome-wide evidence for TF binding, though without TF specificity. In conjunction with a compendium of DNA binding motif models, however, the activity of different TFs can in principle be resolved in terms of global accessibility of the motif matches (i.e. regions whose sequence matches the motif to a reasonable degree). A number of methods from transcriptomics can be repurposed for this task [8–10], and a number of others have been developed specifically for it [11–14]. In general, they can be divided into two families: 1) 'fold-change-based' methods do not operate on individual ATAC-seq samples, but on a vector of per-peak fold-changes between conditions (e.g. obtained from differential accessibility analysis), while 2) 'sample-based' methods assign each motif model a score in each sample, which can then, if desired, be compared between experimental groups. To our knowledge, these methods were never thoroughly compared and evaluated.

Here we benchmark methods for differential TF activity inference from ATAC-seq. We curated a set of ATAC-seq datasets with replicates, profiling genome-wide accessibility upon perturbation (activation or repression) of a specific TF with an annotated DNA binding motif model. To limit the effect of indirect downstream effects, which would make the true perturbation (and hence the benchmark) ambiguous, we included only datasets in which the samples were profiled relatively early after the intervention. For ligand-based activation, which is very fast [15], we included only datasets up to 4h after intervention. For CRISPRi knockdowns, which take much longer to take effect, we used a cutoff of 72h. These cutoffs are obviously not perfect and were largely set in a compromise to gain a sufficient number of datasets while

**Table 1. Description of the benchmark datasets.**

| target | perturbation | system | n/group | delay | ref |
|---|---|---|---|---|---|
| NR3C1 | Agonist (Dexa.) | A549 | 3 | 1h | [15] |
| ESR1/2 | Agonist (Estradiol) | MCF-7 | 3 | 45min | [16] |
| NR1H3 | Agonist (GW3965) | Mouse liver | 2 | 4h | [17] |
| NR1H4 | Agonist (GW4064) | Mouse liver | 3 | 4h | [17] |
| GATA1 | CRISPRi KD | K562 | 3 | 72h | [18] |
| GATA2 | CRISPRi KD | K562 | 3 | 72h | [18] |
| RUNX1 | CRISPRi KD | K562 | 3 | 72h | [18] |
| RUNX2 | CRISPRi KD | K562 | 3 | 72h | [18] |
| KLF1 | CRISPRi KD | K562 | 3 | 72h | [18] |
| MYC | CRISPRi KD | K562 | 3 | 72h | [18] |
| BANP | dTAG KD | mouse ESC | 2 | 2h | [19] |

mitigating the impact of indirect effects. The included datasets are shown in Table 1, and the magnitude of peak-level changes in accessibility upon treatment are illustrated in Fig A in S1 Appendix. We further complemented these real datasets with semi-simulations incorporating perturbations of only one specific TF at a time using ChIP-seq derived peak coordinates.

# 2 Materials and methods

## 2.1 Datasets preprocessing

For the baseline data for simulation, as well as for the NR3C1 benchmark dataset, filtered alignments and peaks were downloaded from ENCODE (NR3C1: ENCSR385LRX; for the datasets related to the simulations, see Table 2).

**Table 2. Data used for the semi-simulations.**

| Database | ID | Purpose | Cell line | ref |
|---|---|---|---|---|
| ENCODE | ENCFF495DQP ENCFF130DND ENCFF447ZRG ENCFF966ELR ENCFF358GWK ENCFF963YZH | Baseline ATAC-seq samples | Human lymphoblastoid cell lines | [36] |
| GEO | GSE98477 | Log fold-change distribution (haploinsufficiency) | Lymphoblastoid cell lines | [38] |
| ENCODE | ENCFF395FIP | Peaks (activation) | A549 | [15] |
| ENCODE | ENCFF416YUN ENCFF860QUX ENCFF456CMP ENCFF181HLP ENCFF870WJP | Reads for log fold-change distribution (activation) | A549 | [15] |
| ENCODE | ENCFF156OCY | CEBPB peaks | GM12878 | [36] |
| ENCODE | ENCFF592UDD | CTCF peaks | GM12878 | [36] |
| ENCODE | ENCFF250FJC | MAZ peaks | GM12878 | [36] |
| ENCODE | ENCFF500EWB | ZNF143 peaks | GM12878 | [36] |
| GEO | GSM3315605 | alternative GC content reference distribution | MCF-7 | [36] |

For the SpearATAC-seq datasets (GSE168851: GATA1, GATA2, RUNX1, RUNX2, KLF1, MYC), the fragment file and processed SummarizedExperiment object were downloaded from GEO. We used the processed object to identify the 3-day cell barcodes assigned to each guide RNA. Then, we merged the fragments for each guide RNA, creating 21 pseudobulk samples (3 guides per 6 target genes + 3 control guide RNAs). For methods that specifically require BAM files, the fragment files were converted using bedToBam.

For other datasets (ESR1/2: GSE117940, NR1H3/4: GSE149075, BANP: GSE155604), and the in-house TRAFTAC dataset, we downloaded the raw reads (if published) from SRA, removed adapters using Trimmomatic 0.38 [20] with ILLUMINACLIP:$adapters:2:15:4:4:true LEADING:20 TRAILING:20 SLIDINGWINDOW:4:15 MINLEN:25, aligned using Bowtie2 2.3.4.3 [21] with –dovetail –no-mixed –no-discordant -I 15 -X 2000 against either GRCh38. p10 or GRCm38. Duplicated fragments were marked by using Picard 2.18.27, and peak calling was done using MACS2 2.2.5 using BAMPE (or BEDPE) mode. Peaks overlapping ENCODE blacklisted regions [22] were removed. Consensus peaks were obtained by merging the peak calls of the individual samples (i.e. GenomicRanges::reduce). Since some methods recommend resizing peaks to a common size, we resized peaks (to 300bp, which is recommended by chromVAR and was close to the median peak size) for all methods, and then generated peak count tables using 'chromVAR::getCounts'.

For nucleosome-free filtering, we used fragments between 30 and 120bp. Given the reduction in effective number of reads, we called peaks with MACS2 on the merged fragments across samples, using a q-value threshold of 0.01. We then resized peaks to 150bp (again because this was very close to the median peak size).

Peak-level differential analysis was performed with edgeR 3.32.1 [23].

## 2.2 Motif models and motif matching

For all methods, we used the HOCOMOCO 10–11 motif models [24] available in the MotifDb package version 1.32.0 [25]. Specifically, we queried for HOCOMOCO motif models of the respective species in the database and kept the newest and highest-quality motif for each gene symbol (see the 'getNonRedundantMotifs()' function of the repository). The BANP motif model was not available in any database and therefore, we obtained the published one [19] from the authors and added it manually to the collection. Similarly, NR1H3 had a human motif model, but no mouse model; we therefore complemented the mouse motif collection with the human one.

To avoid this being a source of variation between methods, whenever possible we tried to standardize the motif matching across methods, using a common fimo 4.11.2 [26] motif scan with default parameters. monaLisa, however, did not allow us to provide the motif matches and performed its own motif matching. While it is a little more stringent, its global number of motif matches was also within 10–20% different of the fimo matches, indicating that it is nevertheless fairly similar.

**2.2.1 Clustering of motif models.** For each species, the motif models were compared with the universalmotif package, using the pearson correlation of the Position Probability Matrices with positions weighted by total information content. Hierarchical clustering with complete linkage was performed on the distances (1—correlation), and the dendrogram was cut at a height (0.35 for mouse and 0.45 for human, which has more motifs) established so that the merge of the clustered motifs did not have a correlation above 0.7. Motif merging was done with the ALLR method of the universalmotif package. The clustered and merged motifs are henceforth referred to as 'archetypes' (similar to [27]).

## 2.3 Differential motif activity methods and their application

**2.3.1 chromVAR.**   While originally designed to analyze sparse chromatin-accessibility data from scATAC-seq data, chromVAR [11] is also commonly applied to bulk ATAC-seq data. The algorithm computes deviations between the summed fragment counts of peaks containing a match for a certain motif model and the expected number of fragments based on the average of all samples/cells, and then compares these deviations to that of random sets of (background) peaks with similar GC content and average accessibility as those containing motif matches, to adjust for technical biases. Specifically, the method produces two matrices of sample/cell- and motif-specific scores: bias-adjusted deviations, and z-scores compared to a background. These scores can then be used to perform differential analysis across groups of cells/samples. The package includes a function to this effect, 'chromVAR::differentialDeviations'. In addition, we tested a variation using 'limma::eBayes' [28] to test for differences in chromVAR activity scores between groups, either based on the adjusted deviations (denoted 'chromVAR(deviations)>limma') or on the z-scores (denoted 'chromVAR(z)>limma'), as well as different normalizations of these scores before differential analysis. Importantly, we observed that the results of the differential analysis were not reproducible across different random seeds unless we considerably increased the default number of background sets (Fig Ba in S1 Appendix). All comparisons were therefore run with 2000 background iterations. Otherwise, the package's guidelines were followed.

**2.3.2 monaLisa: Binned motif enrichment.**   MonaLisa [12] comes with two distinct algorithms, and the first follows a binned enrichment analysis approach. First, ATAC-seq peaks are grouped into bins according to their change in accessibility (i.e. log-fold-change, or logFC) across conditions. Over-representation of motif matches is then assessed for each bin, either in comparison to the bin centered around a logFC of zero, i.e. the peaks not changing (this variant is here denoted as 'monaLisa.vsZero'), or against all other bins (denoted 'monaLisa.vsOthers'). The method can also correct for differences in sequence composition including GC bias in the same fashion as HOMER [29]. In some rare cases, the binning procedure triggered an error due to ties in the logFCs; to avoid this, we therefore added a very small amount of noise to the logFC vector (jitter with a factor of 0.001). monaLisa produces a p-value for each bin and motif model, which is not easily comparable to alternative methods that produce a single p-value per motif. Since the bins' p-values cannot be deemed independent, we tested different p-value aggregation methods that did not assume so, including Simes, Cauchy [30], mean and geometric mean p-values and settled for Simes' as offering the best results in terms of both the rank of the true TF and the network score in the benchmark datasets (see metrics and ranking sections below).

**2.3.3 monaLisa: Randomized Lasso stability selection.**   The second method included in monaLisa follows a randomized lasso stability selection. The goal here is to select TFs as predictors for the observed changes in accessibility. Due to the nature of the robust regression approach, similar motifs compete to be selected. To correct for GC bias, the GC content of the peak sequences can be included as an additional predictor. The model returns a selection probability for each predictor, which we used as a ranking of the motifs for benchmarking. The selection probability cannot be interpreted as a p-value, and since the authors recommend a threshold of 0.9 for stringent selection, we used this threshold for computing precision and recall.

**2.3.4 diffTF.**   diffTF [13] is another method based on peaks logFC between conditions, but which bins peaks by GC content to be more robust to technical variations. Within each bin, the logFCs of peaks with matches for the given motif are compared to those without. It can be run in two modes: in the sample permutation mode, the differences are compared to

differences between permuted groups of samples, while the analytic mode establishes p-values analytically. The latter is more appropriate for small sample sizes, as is the case in our benchmark datasets. We nevertheless tested both modes, and since (as expected) the analytic mode systematically performed better or equal to the permutation mode, the latter was omitted from the main figures. diffTF was used via the pre-configured Singularity container provided by the authors, following the quick start guide and using a maximum of 8 cores. To make the method comparable with alternatives, we provided the motif matches using the same scans as for other methods. Everything else was kept to the default settings.

**2.3.5 VIPER and msVIPER.** The viper package was especially developed to infer differentially-active TFs from transcriptomic data, using either co-expression networks [10] or curated regulons [31]. While it was not used in the context of ATAC-seq to our knowledge, given the similarity of the tasks it was a natural solution to try, simply by converting motif matches into regulons, which become the sets of peaks containing a match for each motif, along with the motif matching scores (scaled by the maximum observed matching score for each motif model). The 'tfmode' (repression or activation) was set to 1 for all targets. Based on the same regulons, the package offers two main functions: the 'viper' method uses an analytical rank-enrichment analysis to compute per-sample TF activity score, while the 'msviper' method works on the vector of per-peak fold-changes between conditions. For the 'viper' method, we first transformed the expression matrix using DESeq2's variance-stabilizing transformation and then used viper's t-test method, and then used limma on the samples' activity scores. Otherwise, both methods were used with default parameters. We tested both approaches using either regulons weighted (likelihood of the links) by the motif scores (denoted e.g. 'VIPER (scores)'), or with all likelihoods set to 1 (denoted e.g. 'VIPER(binary)').

**2.3.6 decoupleR.** decoupleR is a generic package to extract, from -omics data (typically transcriptomics or proteomics), what it conceptualizes as 'biological activities'—of transcription factors, pathways, or enzymes, etc [9]. It includes a unified interface to use a variety of methods, as well as an ensemble ('consensus') method. We tested the top methods from the decoupleR paper: univariate linear models (ulm), multivariate linear models (mlm), weighted sum (wsum), univariate decision tree (udt), as well as the consensus method. Of note, decoupleR can in principle be applied to the peak logFCs, yielding direct significance estimates of differential activity, or (the typical workflow) to the normalized peak count matrix, yielding motif activity scores per sample. We tested both approaches, and for the second (denoted here 'decoupleR>limma') once again applied limma::eBayes to perform differential activity analysis on the motif activity scores.

**2.3.7 fGSEA.** Geneset Enrichment Analysis (GSEA) is a powerful way to look for enrichment of certain sets (in this case, sets of peaks containing a match for each motif model) in a vector of scores per element (in this case, peaks' logFC between conditions). To circumvent the long running time of traditional GSEA, we used the fast multi-level implementation from the fgsea package [8]. To avoid ties in the results, we set nPermSimple = 10000 and eps = 0.

**2.3.8 ATACseqTFEA.** ATACseqTFEA [32] works under the premise that TFs leave a footprint, and calculate, for each site, a weighted binding score that takes the insertion counts proximal to the motif match in relation to those at the matching sequence itself, weighted by the distal count. It then computes the difference in mean scores between the two experimental groups for each site and performs a GSEA-like enrichment score across the ranked peaks for all sets of peaks containing a match for each motif model. The method was run on motif matches inside peaks, as recommended.

**2.3.9 BagFootLike.** The original BagFoot method [14] does not involve proper statistical testing, is bound not to work for the many factors that do not leave a footprint [33], and furthermore we were unable to prepare the various annotation files needed by the software for

the same genome used by other methods. For these reasons, we implemented a logic inspired by BagFoot, again using limma to test for differences in both the footprint depth and the flanking accessibility, and then combining those tests using Fisher's method. As in the original Bag-Foot, we used 20bp on each flank of the motif matches to calculate footprint depth, while flanking accessibility was calculated on the 200bp on each side. We left out the sequence bias correction (reasoning that it is the same across samples, and should not impact relative analyses).

**2.3.10 Insertion model.**   Inspired by various methods [14, 32] and the fact that, when hypothesizing a difference in motif activity, one commonly compares the aggregated insertion profiles around motif matches, we developed a method that counts the (shifted) insertion events around (+/- 200bp) motif matches overlapping peaks for each sample, then for each motif model, weights the insertions based on their position to the motif match. The weights are based on the global insertion profile of the motif model across samples, smoothed and made symmetrical (see for example Fig Bb in S1 Appendix). The weighted insertion counts are then summed to produce a score per motif and sample, and differential motif activity is done using limma.

**2.3.11 MEIRLOP.**   Motif Enrichment In Ranked Lists of Peaks (MEIRLOP) performs a logistic regression of whether regions contain a motif on given region scores (in our case, accessibility log2FC across conditions) and sequence-based covariates [34].

**2.3.12 Simple linear regression approaches.**   We also tested simple per-motif linear regression of the logFCs on either 1) whether or not the peak contains a match for the motif ('ulm(binary)'), 2) the motif matching scores ('ulm(scores)'), as well as 3) whether or not the peak contains a match plus the peak's GC content as a covariate ('ulm+GC').

Finally, we tested a lasso- or ridge-regularized linear regression of the logFCs on the matrix of (binary) motif matches.

As we tested multiple method variations, the short names referring to each and their description can be found in S1 Table.

## 2.4 Benchmark metrics

For rank-based metrics, the results of each method were sorted first according to uncorrected p-values, then by absolute effect size (e.g. absolute logFC, coefficient or enrichment score). This was especially important for permutation-based methods that often produce ties.

To compute the rank of the true motif (e.g. Fig 1), we considered the true motif model that of the manipulated TF, with some special cases: the estradiol treatment is expected to affect both estrogen receptors (ESR1 and ESR2), and therefore both were considered as true; as far as is known, MYC only binds to DNA in cells when in a heterodimer with MAX, and since the MYC motif model was not detected by any method but the MAX was by the top methods, we considered both as 'true motifs'; similarly, NR1H4 (also known as FXR) typically binds DNA as a heterodimer with RXR, and were therefore included also RXRA and RXRB as true motifs. For datasets that had more than one true motif, we used the best rank of any true motif per method as the 'rank of the true motif'.

For the 'network score', we first downloaded from BioGRID 4.4 all the known physical interactors of each true TF (in the species of the respective dataset), and any motif model that overlapped with these interactors (as well as the true motif itself) was considered a network member. For each method, and for k = 1:100, we calculated the proportion of top k motif models that are network members, producing a curve as shown in Fig Bc in S1 Appendix. We then computed the area under this curve (AUC) and divided it by the maximum AUC theoretically obtainable for the dataset (which depends on the number of annotated interactors). The

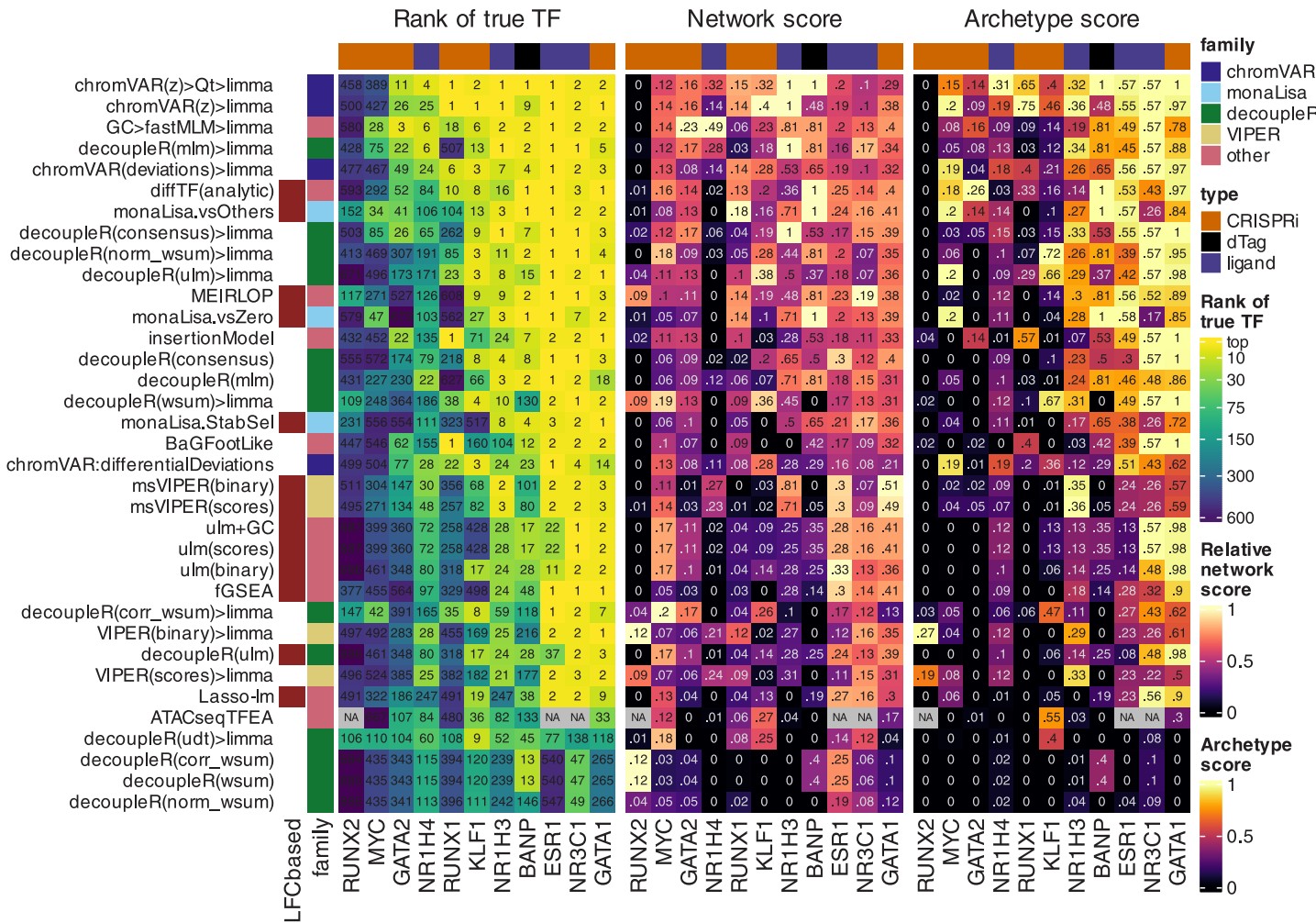

**Fig 1. Benchmark of differential TF activity methods.** Ranking-based metrics for each method across each dataset. The left part of the heatmap indicates the rank of the true TF (top one if the dataset has more than one true target), the numbers are the actual ranks and the colors are mapped to the squared root of the ranks (because differences in very high ranks are irrelevant in practice). The central part shows the network score, i.e. the AUC of the proportion of top X TFs that are in the network of interactors of the true TF, relative to the maximal possible AUC for the dataset (see Methods and Fig Bc in S1 Appendix for details). The labels indicate the actual scores, while for the color mapping, the values are relative to the maximum achieved value for each dataset. Finally, the right part indicates a similar AUC-based score using motif models that are similar to (i.e. cluster together with) the true motif (see Methods). The methods are ranked based on the average of transformations of the three metrics across datasets (see Methods). For ATACseqTFEA, NA values indicate that the method repeatedly crashed on those datasets, presumably due to unsustainably high (>70GB) memory usage. For methods for which variations were tested, we show here only the original and top variant(s). All variants are available in Fig E in S1 Appendix. S1 Table contains full descriptions for the short names displayed here.

'archetype score' was computed in exactly the same manner, but using (instead of interactors) motif models that clustered together with the true motif (see motif clustering above).

Precision and recall (e.g. Fig 2) were calculated using an adjusted p-value <= 0.05, except for monaLisa's stability selection method (which does not provide p-values), for which we used a selection probability of 0.9 as the threshold. Otherwise, if the methods did not report an adjusted p-value, one was generated using the FDR method [35]. To calculate precision, all motifs clustering with network members were counted as positives. However, since not all network members can be expected to be relevant in a given cellular context, only the aforementioned 'true motifs' were considered for calculating recall (i.e. sensitivity).

**2.4.1 Ranking.** For ranking the methods in the heatmaps (e.g. Figs 1 and 3), we transformed the ranks of the true motif using an inverse-logit-based transformation to put it on a

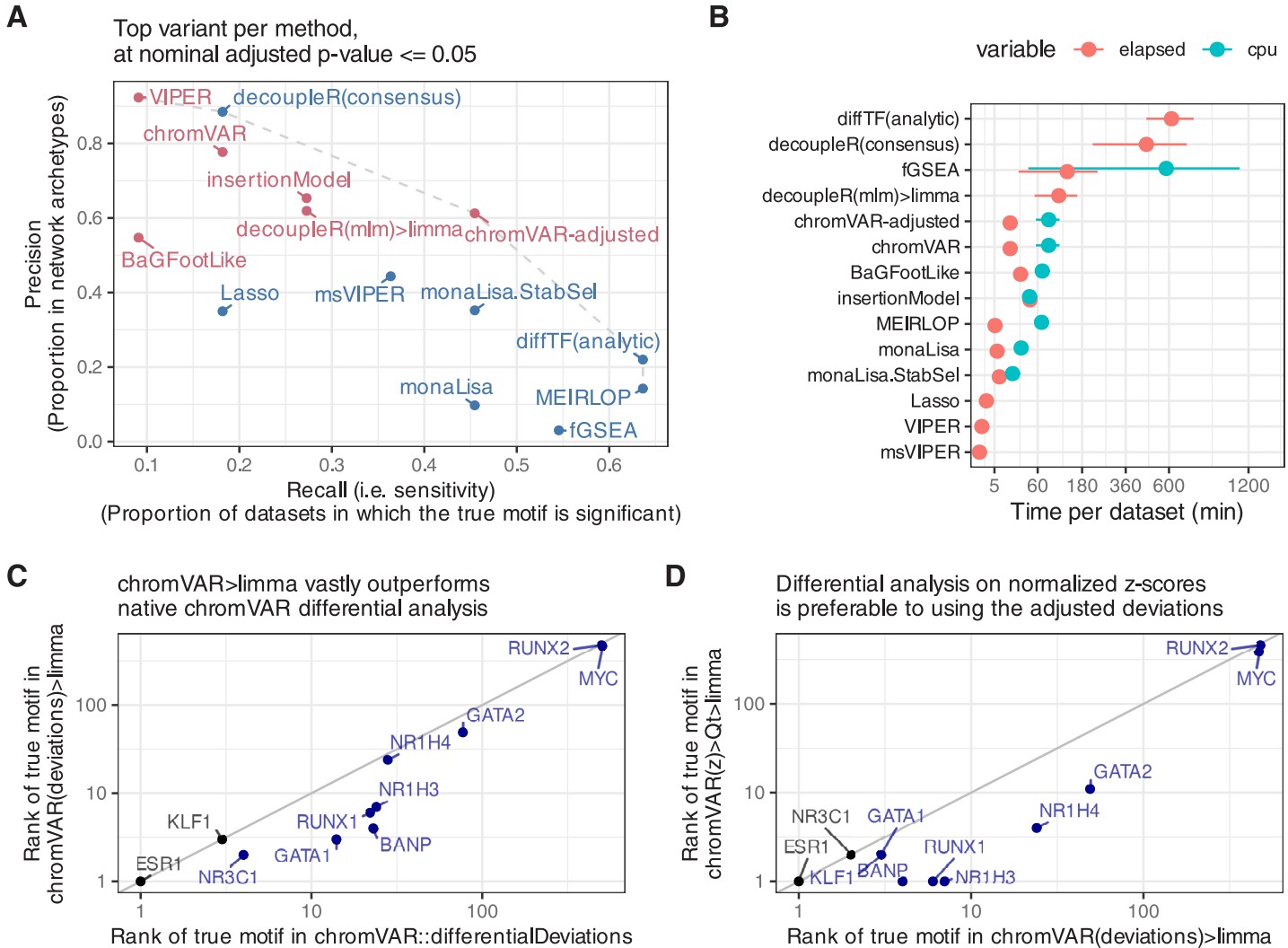

**Fig 2. Further benchmark results, and improvements over native chromVAR. A.** Sensitivity and specificity (considering interactors and members of the same archetypes as positives) of the top alternative per approach. The methods are coloured by family (sample-wise methods in pink, logFC-based methods in blue). **B.** Mean (and standard error of the mean) running time (elapsed, as well as total CPU time when multithreading) across datasets of the top alternative per approach (the x axis is squared-root-transformed for visibility). Note that because it was done separately for reasons of standardization, the running time does not include the generation of the peak-count matrix, nor, except for monaLisa, the motif scanning. monaLisa is therefore disadvantaged in this comparison, and these times should be interpreted as rough estimates. **C.** Comparison of the rank of the true motifs obtained by a *limma* analysis on the chromVAR deviations, versus using chromVAR's native *differentialDeviations*. **D.** Comparison of the rank of the true motifs obtained by a *limma* analysis on the normalized chromVAR z-scores, versus on the chromVAR deviations.

similar scale as the other metrics, and so that it decays beyond a certain point. Specifically, we used:

$$2\sqrt{\frac{e^{-x}}{(1 + e^{-x})}} \tag{1}$$

where $x$ is the square-root of the rank. We then ranked according to the mean across all scores. For dataset/method combinations that had missing values, we imputed the values for the purpose of ranking by taking the median or mean for that dataset, whichever was the worst.

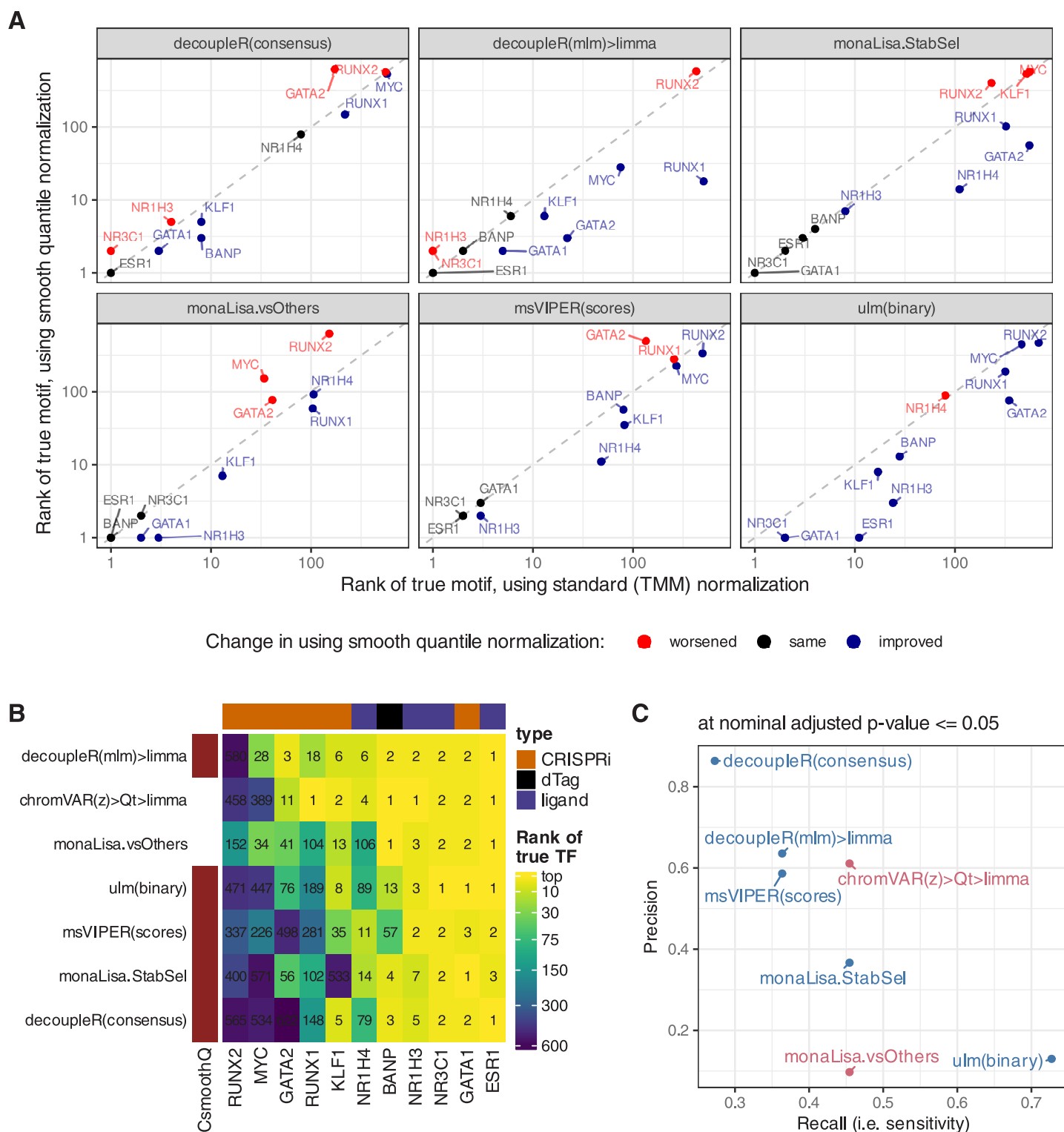

**Fig 3. Impact of using GC smooth quantile normalization. A.** Comparison, across key methods, of the ranks of the true motif when using GC smooth quantile normalization instead of standard (TMM) normalization to calculate per-peak log(fold-change). **B.** Comparison of the ranks of the true motif with GC smooth quantile normalization, compared to the two best-performing methods from the earlier benchmark. **C.** Comparison of the precision and recall (at adjusted p-value <= 0.05) when using GC smooth quantile normalization (in blue), compared to the two best-performing methods (in pink).

### 2.5 Semi-simulations

The semi-simulations were performed by introducing perturbations in the signal of one TF at a time in a baseline ATAC-seq dataset.

As a baseline, six (control) samples of a lymphoblastoid cell line (HG03442) dataset were downloaded from ENCODE [36] (see Table 2 for an overview of the data used for the semi-simulations). To obtain specific perturbations, the baseline ATAC-seq datasets were overlapped with ChIP-seq peaks of that TF from the same cell type. The baseline ATAC-seq samples were divided into two groups, and peak-overlapping regions were downsampled in one of the groups (see Fig J in S1 Appendix).

**2.5.1 Fold-changes.** The extent of downsampling followed one of two differential binding scenarios (TF activation and haploinsufficiency), each based on a real reference ChIP-seq dataset (Table 2) in which the distribution of log-fold-changes between conditions is associated to the degree of respective peaks' enrichment over the input (Fig 4B). Specifically, the enrichment of the peaks to be downsampled was quantile normalized to those of the reference peaks in order to select the fold-change of the closest matching reference peak. The matched per-peak fold-changes ($FC_i$) were introduced in the baseline ATAC-seq samples by downsampling in one of the groups ATAC-seq fragments overlapping the corresponding ChIP-seq peaks (Fig 4A). To test sensitivity of the methods to the strength of the perturbations, the fold-changes were further multiplied by a factor $p \in \{0, 0.25, 0.5, 1, 3\}$, that we refer to as 'perturbation strength'. For a sample $j$ of the group with a perturbation to be introduced for peak $i$, the number of fragments to sample $n_{ij}^s$ was finally obtained by:

$$n_{ij}^s = \min\left\{ \lceil \frac{n_{ij}^0}{FC_i * p} \rceil, n_{ij}^0 \right\} \tag{2}$$

where $n_{ij}^0$ denotes the number of fragments overlapping peak $i$ in sample $j$ in the original dataset. The group in which the fragments overlapping peak $i$ were downsampled, was determined by the sign of the $\log_2(FC_i)$. A perturbation strength of 0 corresponds to the original dataset without any downsampling of ATAC-seq fragments in ChIP-seq peak regions.

Of note, the semi-simulations rest on the assumption that changes in binding are associated to corresponding changes in accessibility, which is not always the case.

**2.5.2 Technical variation.** Further, technical variations were inserted in some semi-simulations by varying GC content and fragment length (FL) distributions (see Fig Jc in S1 Appendix for an overview of the semi-simulated datasets). Fragment GC content distributions were varied sample-wise by sampling fragments according to their GC content, mimicking reference GC content distributions. FL distributions of the baseline ATAC-seq samples were varied by sampling nucleosome-free-, mono-, di- and multi-nucleosome (>2) containing fragments with differing sampling probabilities. Fragments were classified by applying fixed thresholds on their lengths ((0, 120], (120, 300], (300, 500], (500, ∞)).

After obtaining the semi-simulated ATAC-seq profiles by (under-)sampling fragments in ChIP-seq peak regions or according to their GC content or length, peaks were called for each of the groups separately using MACS2 and the peaks of both groups were merged using the bedtools [37] merge command.

### 2.6 TRAFTAC experiment

**2.6.1 Cell culture.** HEK293 cells were cultured in Dulbecco's Modified Eagles Medium (DMEM) with 10% Fetal Bovine serum (FBS) (Gibco,10500064) and 1% Penicillin/Streptomycin (P/S) (Gibco,15140122). All of the incubation steps were performed at 37˚C and CO2

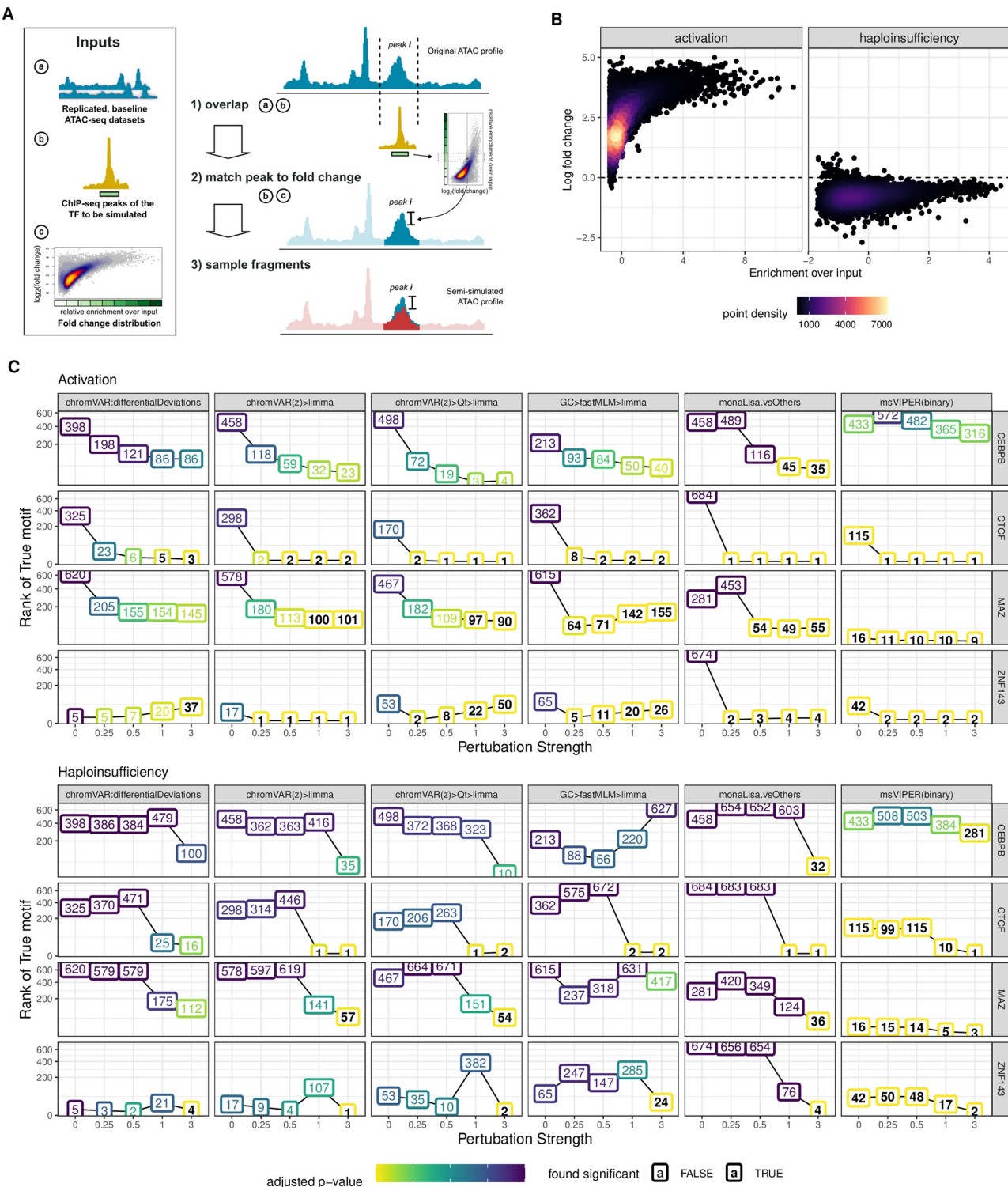

**Fig 4. Semi-simulations and impact of perturbation strength. A.** Strategy for sampling ATAC-seq fragments: ChIP-seq peaks of the TF to be perturbed are overlapped with the baseline ATAC-seq profiles (1); a fold-change is determined by per ChIP-seq peak fold changes from a reference distribution (2) and used to sub-sample the overlapping fragments of the ATAC-seq profiles of one of the two groups (3). **B.** Two group designs were simulated by downsampling ATAC-seq fragments in one group according to the obtained fold changes of the overlapping peaks. Perturbation strength was varied by multiplying fold changes with different factors (0, 0.25, 0.5, 1, 3), i.e. the perturbation strength. Further technical variations were introduced in some datasets by varying GC contents and fragment length distributions. **C.** Performance of a subset of the top performing methods on

the semi-simulated datasets with varying perturbation strength. Numbers in the line plot represent the rank of the perturbed TF as detected by the respective method. Colors of the boxes signify the corresponding adjusted p-value obtained in the differential activity analysis. Bold vs plain text indicates if the adjusted p-value was found to be significant (adjusted p-value $<= 0.05$).

content of 5%. For the transfection cells at a confluency of 70–90% were incubated for 6 h with 1 $\mu$g/mL of a plasmid encoding N-terminal HA-HaloTag7-containing dCas9 fusion protein and 25 nM TRAFTAC using Lipofectamine 3000 following the suppliers' instructions. The plasmid used was published and kindly provided by the Crews lab [39]. Cells were then incubated for 19h with 5 ng/mL of TNF-$\alpha$ (Sigma-Aldrich, H8916) and 25 $\mu$M HP14 (dissolved in DMSO) (WuXiAppTec) respectively DMSO. 30 min before harvest, cells were treated with 200 units/ml DNase (Sigma, DN25). HEK293 cells were then washed twice with PBS (Gibco, 10010023) and detached with TrypLE (Gibco, 12605010). Enzymatic digestion was quenched with growth medium and cells were washed once again with PBS.

**2.6.2 Protein detection.** Western blot was performed to qualitatively and quantitatively assess NFkB levels. Overall protein abundance was determined via BCA assay following the ThermoFisher scientific protocol. The obtained values were normalized and utilized to calculate the sample volumes to be loaded. Samples were mixed with Laemmli buffer (BioRad, #1610747) and $\beta$-Mercaptoethanol (Sigma, M6250) (1:10) and boiled for 10 min at 70˚C. SDS-PAGE was performed using 10% Mini-PROTEAN TGX precast gels and the proteins were transferred from the gel to a Trans-Blot Turbo Mini nitrocellulose membrane via electroblotting. Membranes were then blocked in 5% milk in TBST (1 x TBS, 0.1% Tween 20, pH = 7.6) for 1 h and incubated overnight in primary antibody against NFkB (SantaCruz, sc-8414) (1:200 diluted in 5% milk in TBST), HA (Cell signalling, 3724) (1:1000 diluted in 5% milk in TBST) and GAPDH (Millipore, ABS16) (1:1000 diluted in 5% milk in TBST) at 5˚C. The next day the membranes were incubated for 1 h in HRP-coupled secondary antibody (Sigma Aldrich, AP308P) (1:20 000 diluted in 5% milk in TBST). Proteins were then visualized via chemiluminescent detection (Clarity Western ECL Substrate) at a BioRad imager.

**2.6.3 Library preparation for ATAC-sequencing.** ATAC libraries were prepared from 50 000 cells per sample according to the published OmniATAC protocol [6] with minor adaptations. In short, samples were washed with cold 50 $\mu$l of PBS, afterwards cells were incubated in 50 $\mu$l lysis buffer (10 mM Tris-HCl pH 7.4, 10 mM NaCl, 3 mM MgCl2, 0.1% NP-40, 0.1% Tween-20, 0.01% Digitonin) for 3 min on ice. Cells were washed with 1 ml of lysis buffer without Digitonin and NP-40. Transposition was done in 50 $\mu$l reaction buffer (1x tagmentation buffer (Diagenode, C01019043), 0.1% Tween-20, 0.01% and 2.5 $\mu$l Tn5 (Diagenode, C01070012)) for 30 min at 37˚C and 1000 RPM shaking. Subsequent DNA purification was performed with MinElute Reaction Cleanup Kit (Qiagen, 28204). PCR amplification was done with NEBNext High Fidelity 2X PCR Master mix (NEB, M0541) and UDI primer (Diagenode, #C01011034) with concentrations according to manufacturer's recommendation. Total number of needed amplification cycles (8x) was controlled with qPCR. Library purification and fragment size selection was done with AMPure XP beads (Beckmann, A63880). 20 $\mu$l of 10 nM was sent for sequencing externally with Novogene (Cambridge, UK). A minimum of 20 Mio PE reads were sequenced per sample. The data is available under the GEO accession GSE260504.

## 2.7 Packages versions and code

We used the following package versions: chromVAR 1.12.0, monaLisa 1.5.0, ATACseqTFEA 1.5.0, viper 1.24.0, decoupleR 2.3.3, fgsea 1.16.0, diffTF 1.8.

The code used to run the methods and simulations, evaluate the results, and create the figures is available on https://github.com/ETHZ-INS/DTFAB.

The semi-simulated datasets can be retrieved from Zenodo: 10.5281/zenodo.10732704, 10.5281/zenodo.10781849, 10.5281/zenodo.10781759, 10.5281/zenodo.10781109.

## 3 Results

### 3.1 The top methods were specifically designed for chromatin data

Fig 1 shows the results of the methods on the benchmark datasets. A good method for identifying differentially accessible TF should rank the motif model of the perturbed TF (henceforth 'true motif') top, i.e. with the smallest p-value. We therefore first monitored the rank that each method attributed to the true motif (Fig 1, left). Of note, the MYC and NR1H4 motifs were not accurately detected by any method in their corresponding datasets, but their obligatory binding partners (respectively MAX and RXRA/B) were. These were therefore included in the set of true motifs for the benchmark. The top performing methods were all methods specifically designed for chromatin data, in particular, chromVAR (adapted, see below) and mona-Lisa. Proven methods for geneset enrichment analysis (GSEA) as well as methods with demonstrated performance for transcriptome-based master regulator analysis, such as decoupleR- and viper-based methods [9, 31], underperformed.

Of note, the two datasets in which none of the methods ranked the true motif in the top 10, RUNX2 and MYC, both showed no peak-level change in accessibility with genome-wide significance (Fig Aa in S1 Appendix), suggesting the lack of a clear treatment effect in the first place. However, for some other datasets that showed no significant peak-level change (e.g., RUNX1 and GATA2), some methods ranked the true motif towards the top (though not with an adjusted p-value <= 0.05), thus indicating that global differences in motif accessibility can be detected even in the absence of significant peak-level changes. In the case of RUNX2, it should additionally be noted that the RNA expression of the gene in that cell type is very low, and hence perhaps the factor had a low activity to begin with.

TFs do not act in isolation, but most often form heterodimers or complexes that cooperatively regulate gene expression. Therefore, perturbations on one TF will likely disrupt whole complexes and the binding of its co-factors as well. We therefore devised a 'network score', based on the ranking of known physical interactors of the perturbed TF (see Methods and Fig Bc in S1 Appendix), shown in the middle part of Fig 1. This second metric confirms the findings of the first: except at very low (1–3) or high (>100) ranks (i.e. where ranks cease to be discriminatory), the two metrics show a good agreement, although the network score is not comparable across datasets (Fig Bd in S1 Appendix). Finally, some motif models in the collection are highly similar to the true motif, and these should also tend to be ranked towards the top. To assess this, we clustered similar motif models into archetypes (see Methods), and then computed a score similar to the network score, but based on motif models clustering together. This is shown in the rightmost heatmap of Fig 1, and also largely recapitulates the rank-based relative performances.

Ranking the true motif (and that of its interactors) top is critical, but it does not assess whether the significance of the differential activity reported by a method is well calibrated. We therefore measured the sensitivity in detecting the true TF (recall) in relation to precision. Fig 2A reports those values for the top variant of each method. To calculate precision, all motif models clustering with the true TF or its known interactors were considered as true positives (considering only interactors yielded very similar results). As expected, we observed a trade-off between precision and recall, with fGSEA offering the highest sensitivity but at the cost of a very high FDR, while VIPER, for example, controlled FDR but had a very low sensitivity (Fig

2A). As with the rank-based metrics, our adjusted chromVAR method (chromVAR>Qt>limma) showed the best trade-off between precision and recall.

While monaLisa (without stability selection) performed well in ranking the true motifs (Fig 1), it had a low precision, indicating that it should be used for discovery rather than confirmatory purposes, or in conjunction with the stability selection strategy. That said, it should be noted that monaLisa reports a p-value for each motif model and fold-change bin, and that while our use of Simes' aggregation to obtain a single p-value per motif (needed for benchmarking purposes) is not unreasonable, it does not come from the method's authors.

The estimated mean runtime per dataset for each method is shown in Fig 2B. Despite multi-threading (see Methods), diffTF had very long runtimes; fGSEA, ATACseqTFEA and decoupleR also had elevated runtimes, taking several hours per dataset, while all other methods performed within a reasonable time.

## 3.2 Improvements to chromVAR-based approaches

The chromVAR package includes a function to perform differential analysis ('chromVAR::differentialDeviations'), which performs t-tests on the adjusted deviation scores of each motif. However, this method did not perform well, and we found that using limma [28] on chromVAR's motif z-scores ('chromVAR(z)>limma') considerably outperformed this original approach (Figs 1 and 2). Presumably, this is because of limma's moderation of the variance. In addition, we observed that in some datasets, the per-sample distributions of z-scores could exhibit major shifts and differences in width that were not associated with experimental groups (Fig Ca-b in S1 Appendix). In particular, not resizing peaks to a common size (as is recommended) tended to result in large shifts in the distributions, but some variations in the distributions—especially in their width—persisted despite resizing (Fig Ca in S1 Appendix).

Since these distributional differences were in our experience sometimes correlated to technical sources of variation such as sequencing depth, and are unlikely to have a biological meaning, we hypothesized that something in the chromVAR method did not perfectly adjust for technical differences between the samples. Internally, chromVAR uses a simple library size normalization to compare each sample to expectations, which is known often not to be adequate for NGS data [40]. We therefore hypothesized that these shifts could be due to inadequate normalization, and adapted chromVAR to use more robust normalization procedures, including the well-established TMM normalization [40]. However, these changes had little impact on the distributional differences, nor the differential TF identification (Fig Cc-d in S1 Appendix).

We therefore tested whether a further, downstream normalization of the motifs z-scores would improve the detection accuracy. Quantile normalization appears to perform best (Fig 1 and Fig Da in S1 Appendix). This was also confirmed by comparing the *t*-value attributed to the motif of the true TF, which is more sensitive (Fig Db in S1 Appendix).

In light of these results, we took advantage of the fact that some of the benchmark datasets are in fact (aggregated) single-cell ATAC-seq datasets [18] to see whether such normalization also had a beneficial impact at the single-cell level. As a proxy for the signal-to-noise ratio, we computed the standardized mean difference of the respective motif's z-scores upon each knockdown. Centering the z-score distributions was always beneficial, quantile normalization also tended to be beneficial most of the time, while unit-variance scaling showed mixed results (Fig Dc in S1 Appendix).

Finally, we also tried to optimize another of the top methods, monaLisa. Changing the number of bins or the logFC threshold did not lead to improvements (Fig E in S1 Appendix). Since observing a dose-response across the monaLisa bins is a further evidence of a pattern, we

also tried to capture this computationally by dividing, for the purpose of ranking, the aggregated p-values by the absolute Spearman correlation between the fold-enrichment and the bins order (denoted 'monaLisa.vsOthers+spearman' in Fig E in S1 Appendix), however it did not yield a clear improvement.

### 3.3 Using GC smooth quantile normalization improves non-specialized methods

Sample-specific GC bias is a known source of variation in ATAC-seq, and methods have recently been developed to address it, in particular, smooth-quantile normalization within GC bins [41]. We therefore tested whether using this normalization, in particular for calculating the peak-level log(fold-change) between conditions, could improve in particular logFC-based and non-specialized methods. As shown in Fig 3A, which compares key methods with and without GC smooth quantile normalization, most methods benefited from this specialized normalization method, vindicating the procedure (see Fig E in S1 Appendix for the full rank-based metrics on all variants). Indeed, even monaLisa stability selection, which already includes some form of control for GC bias, seemed improved. With this normalization, the sample-based decoupleR(mlm)>limma approach even became even more competitive with the top chromVAR variant (Fig 3B and 3C). We therefore re-implemented the method in a more performant fashion, and included it (in combination with GC smooth quantile normalization) in the following analyses as 'GC>fastMLM>limma'.

### 3.4 Impact of stringency in motif matching and using fewer, more distinct motif models

The previous analyses used default thresholds for motif matching. To test the impact of variations in this step, we decreased the motif matching p-value threshold by a factor of ten, i.e. from 1e-4 to 1e-5, leading to an average 82% decrease in global motif matches (for monaLisa, where only a score threshold can be set, we changed the threshold from 10 to 13, leading to a 74% decrease in matches). The impact of these changes on a selection of the top methods was relatively mild, but overall a more stringent motif matching had negative effects (Fig F in S1 Appendix).As previously mentioned, many motif models in the collection are highly similar to each other, and as such will get very similar scores from the majority of activity inference methods that test each motif model separately. Multivariate methods, instead, can be side-tracked by very similar motifs. Given these considerations, we assessed whether the relative performance of a selection of the top methods was altered when using the motif archetypes (i.e. a smaller collections of motif models that are less similar to each other). Overall, using motif archetypes yielded a similar or lower performance, in particular a slightly lower power (Fig G in S1 Appendix).

### 3.5 Focusing on nucleosome-free peaks/fragments yields mixed results

ATAC-seq data has a characteristic, roughly periodic fragment length distribution, where smaller fragments (e.g. <100–125bp) are too small to be wrapped around a nucleosome and are therefore deemed 'nucleosome-free' (NF), while larger peaks correspond to mono-nucleosome-, di-nucleosome-, and eventually tri-nucleosome- containing fragments. Since most TF compete with nucleosomes for binding, and NF fragments are enriched at the core promoter, NF fragments are often thought to be particularly characteristic of TF binding. Therefore, we tested whether using filtering based on fragment size could improve the benchmark results, focusing on the top method for each of the two families of approaches. Specifically, in addition

to the previous results based on all fragments, we tested i) counting only NF fragments, ii) counting all fragments but on peaks called only on NF fragments, as well as iii) counting only NF fragments on NF peaks (i.e. using only NF fragments throughout). While these alternatives did appear beneficial on some datasets, their impact on discriminatory power was not consistent across datasets (Fig H in S1 Appendix). In particular, chromVAR hardly ever benefited from these variants, whereas it sometimes made major differences to monaLisa, for instance with GATA2 being much better ranked when counting NF fragments (Fig H in S1 Appendix). Overall, however, these differences did not generalize into clear recommendations.

### 3.6 Limited gain from simple method aggregation

Reasoning that different approaches could in principle offer complementary advantages, we next investigated whether aggregating the results of the three top-performing methods could yield superior results. We tested common p-value aggregation methods (despite the assumption of independence underlying some of them being violated), as well as a non-parametric, rank-based alternative (Fig I in S1 Appendix). The latter proved best but did not show consistent improvement upon the best single methods. We next tested combining the per-sample motif activity scores produced by chromVAR and by the multivariate model applied on GC smooth quantile normalized counts, before computing differential analysis (Fig I in S1 Appendix). This was always better or equivalent to chromVAR, although the improvements were mild. While it is possible that more sophisticated ensemble methods could yield an improvement, these results suggest a low degree of complementarity between methods.

### 3.7 Semi-simulations confirm the results from real benchmark datasets

Even when focusing on early time-points, the real benchmark datasets have a problem of ambiguity with respect to the truth, as interfering with a factor is likely to impact other interacting factors, as well as indirectly interacting factors competing for binding. To confirm the robustness of our results, we therefore developed a semi-simulation procedure. We generated, from baseline ATAC-seq profiles of lymphoblastoid cell lines, cohorts of two groups of samples with a difference in accessibility at binding sites of a single given TF at a time (Fig 4A; see also Fig J in S1 Appendix and Methods). Briefly, fragments overlapping the ChIP-seq peaks of the given TF were subsampled in one of the two groups following one of two realistic differential binding scenario (Fig 4B): an activation (where the high-occupancy peaks are the most affected), modeled on the ligand-based activation of the glucocorticoid receptor [15], and a haploinsufficiency (where the TF amount is halved, and the remaining TF preferentially goes to high-occupancy peaks, leaving the low-occupancy peaks most affected), modeled on the heterozygous loss of function of YY1 [38]. In the activation paradigm, the difference between conditions affects virtually all sites (and many by a lot), whereas in the haploinsufficiency many peaks are very mildly affected (Fig 4B). Specifically, we simulated various perturbation strengths (i.e. multiplicators of the original differential binding fold-changes) for four factors that had 1) high quality ChIP-seq data for the simulated celltype (lymphoblasts), 2) a clear associated motif model, and 3) represented distinct factor families and properties (Fig K in S1 Appendix).

We first confirmed that the simulated data was distributed similarly to the real data (Fig L in S1 Appendix), and that the effects introduced were as expected (Fig M-N in S1 Appendix), and next ran the methods on each dataset. The results, a subset of which is presented in Fig 4C (full results in Fig O in S1 Appendix), afford the following observations. As expected, stronger perturbation strengths made the task easier, and the haploinsufficiency scenario was systematically harder to detect, owing potentially both to the lower general fold-changes and

comparative sparing of high-occupancy sites (Fig 4B). Contrarily to the real datasets, msVIPER appears to perform very well in the simulations (Fig O in S1 Appendix), however this is chiefly because for two hard datasets, it gives by chance low ranks to the TF already in the absence of any simulated effect, which remains low across the simulations. Aside from this, as with the real datasets limma on the chromVAR z-scores gave the best overall results, followed by mona-Lisa (vsOthers) and MEIRLOP, which provided a good performance so long as the perturbation strength was sufficient. Furthermore, monaLisa (vsOthers) offered greater sensitivity than other methods, being the only approach marking the true factor for all simulated TFs and paradigms as significant and also in some cases at lower perturbation strengths, however at the cost of a generally lower precision (Fig P in S1 Appendix). Once more the original chromVAR differentialDeviations did not perform well, and the multivariate model had mixed, somewhat unstable performance.

Given the controlled setting of the simulations, any difference between the factors necessarily hinges on the relationship between its motif model and its experimental binding sites. The easiest factor was CTCF, which had the highest number of experimental binding sites, but especially the highest proportion of those harboring matches for its motif, as well as the highest enrichment for motif matches in bound (i.e. overlapping a ChIP-seq peak) vs all ATAC-seq peaks (Fig Ka S1 Appendix). On the opposite spectrum, CEBPB was the hardest, with few binding sites, few of which with a motif match. Interestingly, ZNF143 was easier to detect than MAZ, despite having a similar number of sites and a motif model that is less specific to the bound peaks (Fig Ka in S1 Appendix). Closer investigation revealed that a large number of motifs co-occur with MAZ, and in fact show a stronger enrichment in its bound peaks (Fig Kb in S1 Appendix). While it is also the case with ZNF143 that many other motifs are more enriched in its bound peaks, this is chiefly because a part of the ZNF143 peaks show a higher GC content (Fig Kc-d in S1 Appendix), and the fact that the top methods correct for GC presumably enables an accurate detection nonetheless. To confirm the importance of the relationship between motif models and binding site, we created simulations in which only the ChIP-seq peaks containing a motif match for the respective TF were perturbed. As expected, for all TFs but in particular for CEBPB and MAZ performance of all methods improved considerably (Fig N in S1 Appendix).

Finally, the semi-simulation procedure further enabled us to introduce sample-wise variations in two known factors of technical variation: fragment length (FL) distribution (Fig Q in S1 Appendix) and GC content (Fig R in S1 Appendix). The top methods were generally robust to both kinds of variation, despite not correcting for FL biases. However, for some of the biased settings (both GC and FL), monaLisa (vsOthers) detected false positives in datasets were no perturbation was simulated (perturbation strength 0).

In summary, the simulations confirm the main results of the benchmark on real datasets, reveal the top methods to be robust to technical biases, and indicate the relevance of how distinct and informative the motif model is for actual binding.

## 3.8 Application of the methods to characterize TRAFTAC-mediated TF degradation

PROTACs (PROteolysis TArgeting Chimera) are a particularly successful technology that hijacks the cell's endogenous protein degradation machinery (the ubiquitin-proteasome system) and targets it to a protein of interest [42]. However, each PROTAC requires careful chemical design and validation. A recent, attractive derivative technology harnesses the versatility of the CRISPR technology to target and degrade TFs in a more flexible fashion. With TRAFTACs, or TRAnscription Factor TArgeting Chimeras [39], the targeting specificity is not

encoded in the design of the PROTAC molecule, but by a double-stranded DNA containing the target TF's binding motif. This oligo is covalently linked to a single-stranded RNA that functions as a guide to recruit a deactivated Cas9 fused with a HaloTag7, which in turn can be bound by a generic HP14 PROTAC. As a result, binding of the oligo was shown to lead to ubiquitination and subsequent cleavage of the target TF [39].

Using the top differential motif accessibility methods identified above, we sought to characterize the impacts of a validated TRAFTAC against NFkB [39]. Replicating the authors' experiment, we transfected HEK cells with the dCas9-HT7 plasmid and DNA-RNA hybrids (NFkB or scrambled TRAFTAC) for 6h, before treating the cells TNF-$\alpha$ (to trigger NFkB activation) and the PROTAC against the Halo tag (HP14) for 19h. We confirmed the reduction of NFKB1 protein levels using western blot (37% reduction with respect to scrambled control, Fig S in S1 Appendix), and generated ATAC-seq profiles. We then ran differential motif accessibility analyses using chromVAR and monaLisa (Fig 5). Interestingly, although the NFkB motif matches have a weak trend of decreased accessibility, they are far from significance (Fig 5A). Instead, the strongest signal is in motifs related to the AP-1 complex, i.e. JUN/FOS family members (which all have very similar motif models), showing a decreased accessibility in the knockdown according to both the chromVAR and monaLisa analyses. Of note, despite the key role of the AP-1 complex in regulation response to stimulus, JUN/FOS-related motifs were significant in only one of the benchmark dataset (BANP), arguing against a widespread sensitivity to perturbation, and for a role specifically related to NFkB/TNF-$\alpha$. Beside activating NFkB through IKK, TNF-$\alpha$ is expected, through MKK7, to trigger the MAPK/JNK pathway, which in turn activates AP-1 [43]. However, the reduction in AP-1 activity that we observe in the NFkB knockdown suggests a cross-talk between these two downstream signaling pathways.

Fig 5C shows some example significant motifs across the top three methods. Some are consistent: for example, the GC-rich GLIS2 motif appears enriched in a dose-dependent fashion in monaLisa bins that increase in accessibility, and depleted in those that decrease; it is moreover increased in activity in the knockdown according to both chromVAR and GC>MLM>limma. NFE2 and FOSL1 show the same consistency, with the opposite pattern. In contrast, some inferred motifs activities are inconsistent across methods: while both chromVAR and MLM suggest a reduction in NFYA accessibility in the knockdown, which would be consistent with reported links between NFkB and CCAAT elements [44], the monaLisa results shows that the motifs are depleted in both upregulated and downregulated peaks, and instead enriched in the bin of peaks that do not change. JUNB and FOS, instead, show the same pattern as FOSL1 according to both monaLisa and chromVAR, as expected given that the motifs are virtually the same, while MLM shows no clear pattern. This is precisely because the three motifs are largely equivalent: because the multivariate approach regresses accessibility on the combined effect of the different motifs, highly-similar motifs will nevertheless have different coefficients. This is a good illustration of the fact that, whereas most methods benefited from the extensive collection of motif models, multivariate methods should be used with a reduced collection of more distinct motif models (Fig G in S1 Appendix).

## 4 Discussion

A major issue in assessing methods for differential TF activity is the lack of a ground truth, even in datasets with known experimental perturbations. Proteins interact in complex ways that modulate both their stability and activity, so that knocking down or altering the activity of a TF is highly likely to affect the activity of other, interacting TFs. Moreover, TFs typically function in cascades, so that intervening on a TF will affect the transcription, and eventually activity, of other downstream TFs, leading to indirect, yet no less genuine changes in TF

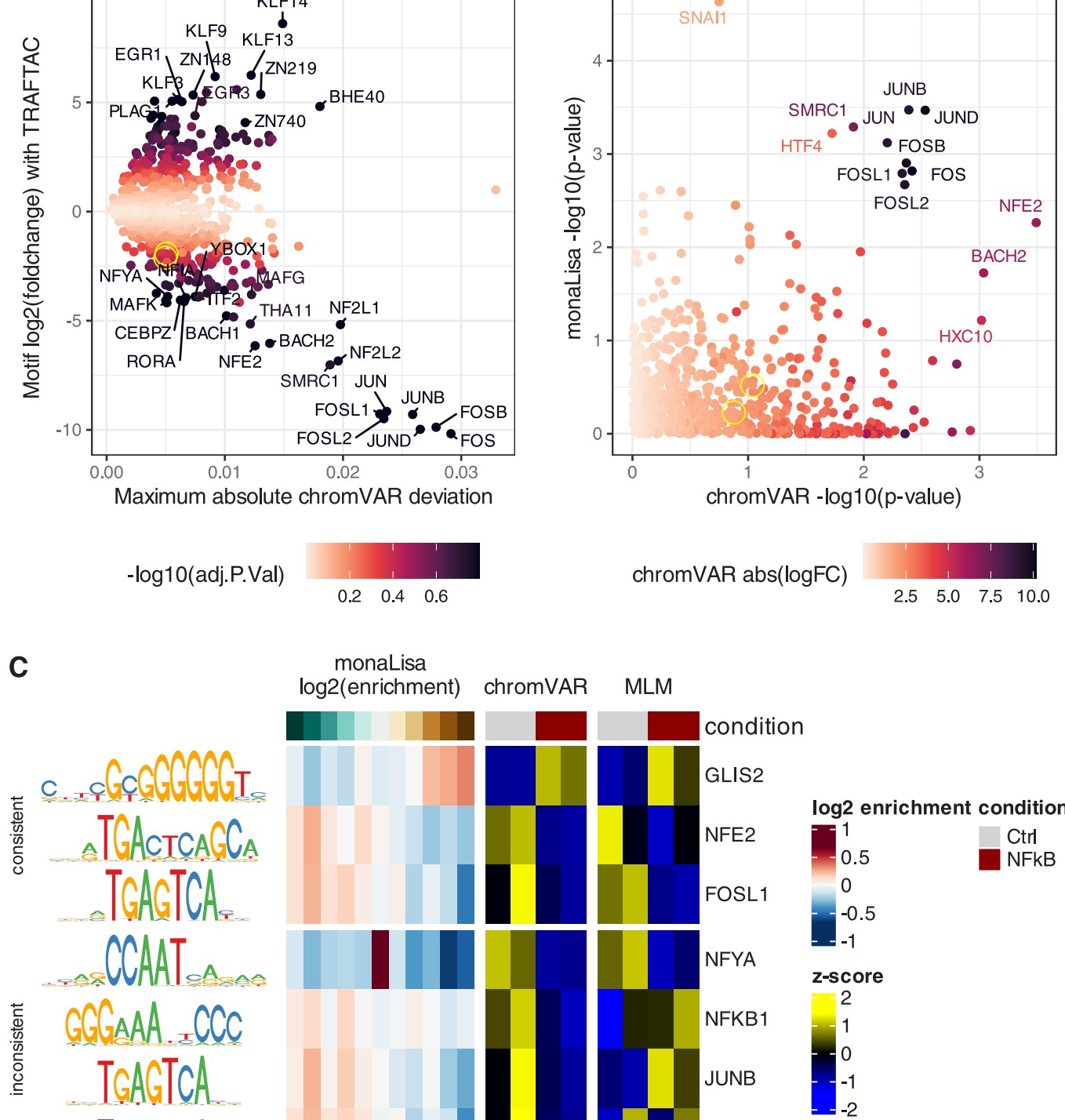

**Fig 5. Characterizing the impacts of a TRAFTAC against NFkB after TNF-α activation in HEK cells. A.** Results of the 'chromVAR(z)>limma' analysis, highlighting the motifs with strongest changes between conditions and high absolute deviations. The two overlapping yellow circles highlight NFKB1 and NFKB2. **B.** Comparison of the per-motif p-values assigned by monaLisa (Simes p-value) and chromVAR, showing an agreement only on the strongest differences. **C.** Representative examples of motifs called as differentially-accessible by some method, showing side by side monaLisa's enrichment scores across logFC bins, as well as the per-sample chromVAR z-scores and z-scores of the MLM t-values.

activity. We adopted a number of complementary strategies to mitigate these issues. First, we focused on datasets profiled shortly after intervention to minimize indirect effects. Second, developed a network score and computed precision in a way that considers known interactors of the perturbed TFs. Finally, we developed a semi-simulation procedure that avoids both issues. While none of these solutions are perfect, the general agreement between simulations and real datasets, as well as between evaluation metrics, provide us with the confidence that our results genuinely reflect the relative performances of the methods.

## 4.1 Summary of the results and general recommendations

Fig 6 summarizes the main benchmark results on the union of the top 5 methods from real and semi-simulated datasets. Overall, our adjusted chromVAR workflow performed best on rank-based metrics, and provided the best tradeoff between sensitivity and specificity. Specifically, we recommend increasing the number of backgrounds iterations (to 1000–2000) to improve reproducibility (not easily done with single-cell data), and using limma on the z-scores to perform differential analysis (as opposed to the native 'differentialDeviations', which did not perform well). Moreover, its results were often (though not always) improved by post-processing of the scores, suggesting that some technical variation is not entirely accounted for by chromVAR's background strategy. Furthermore, we found the sample-wise methods useful

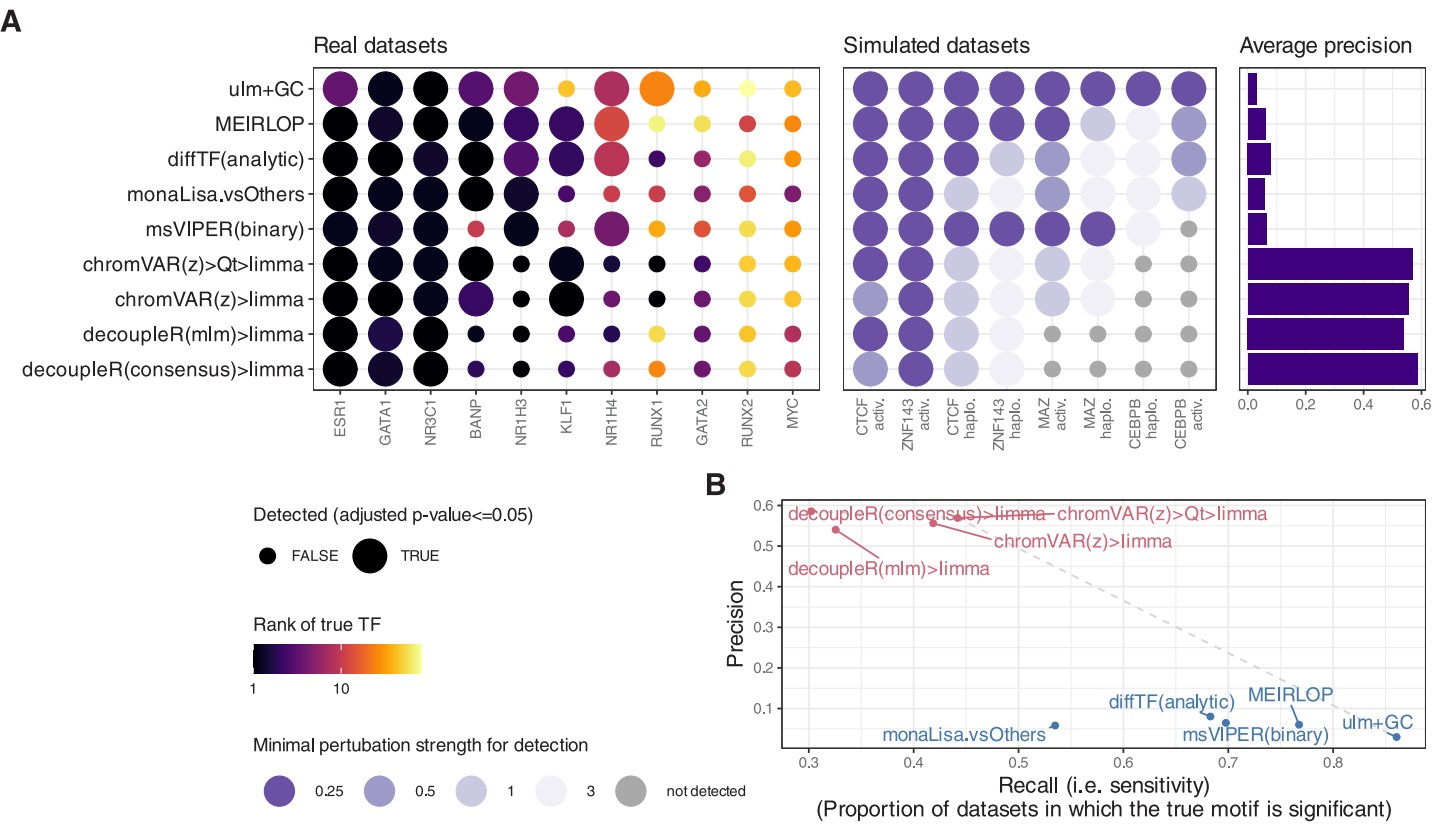

**Fig 6. Summary of the benchmark on variants of the top methods. A.** Sensitivity (left and center) and precision (right) of each method across real (left) and semi-simulated datasets (center). The size of the points indicate whether or not the method found the true motif significant (adjusted p-value <= 0.05). For real datasets, the color indicates the rank of the true motif, while for semi-simulated datasets it indicates the perturbation strength at which the true motif was found significant. **B.** Sensitivity and specificity (considering interactors as positives) of the top methods, aggregating both real and semi-simulated datasets. The methods are coloured by family (sample-wise methods in blue, logFC-based methods in pink).

in relaying to the analyst the variability across samples. On the other hand, those methods are limited to detecting global increases or decreases in TF activity, and in principle cannot detect the displacement of a TF's binding at constant global activity (although it is unclear how likely a scenario this is). MEIRLOP, diffTF and monaLisa also gave good results, although with a low precision. monaLisa's bin-based strategy and plotting functions have the advantage of providing the user with a more fine-grained view of where the enrichment is observed, relative to where the changes in accessibility are happening. Observing a dose-response (i.e. an increased enrichment in bins of greater fold-change) bolsters our confidence in the enrichment.

While methods not specifically designed for chromatin data performed poorly in the main benchmark, they were improved when using smooth quantile normalization in GC bins, confirming the value of the normalization method. In particular, the GC>decoupleR(mlm)> limma approach performed as good as, and in some datasets better than, the top specialized method. This method fits a multivariate linear model on the peak accessibility of each sample, using all motifs as covariates. Of note, it is one of the few methods (along with monaLisa stability selection and the similar lasso-linear model, which are however logFC-based rather than sample-based) that handles motifs together, rather than separately. This suggests that this general strategy should be further investigated. However, as the TRAFTAC example illustrates (Fig 5C), a high motif similarity within the collection causes major problems for multivariate approaches, whether regularized or not. When using such methods, we rather recommend using a reduced collection of more distinct motif models. Most likely, harnessing the potential of the multivariate approach will involve a more refined handling of the similarity between motif models.

Simple results aggregation of these three methods did not yield clear improvements. Nevertheless, we recommend using more than one of the top methods to increase confidence, given their different strategies and benefits. Furthermore, we observed that the p-values were not generally well-calibrated, i.e. the observed false discovery rate was nearly always higher than the nominal threshold, despite not counting similar motifs and interactors as negatives (admittedly, the knowledge of interactors taken into account for calculating precision is neither exhaustive nor specific to the cell types used). Precision estimates show strong differences across datasets, and high precision typically only occurred when no effect was detected (Fig P in S1 Appendix). Hence, we recommend caution in interpreting the obtained precision estimates, and for this reason we especially focused on ranks of the true motif for reporting performance. However, for similar reasons also the ranks should be interpreted on a rather coarse scale, and might not be comparable across TFs given differences in the number of factors co-binding/co-occurring with the true factor.

During the revision of this manuscript, a related study was published evaluating methods for TF prioritization ATAC-seq and, especially, a large number of H3K27ac datasets with known perturbations [45]. Consistent with the present study, they also identified monaLisa as one of the top methods, indicating its robustness to different types of epigenomic signals. On the other hand, they also reported MEIRLOP as a top method, which performed decently but not as well in our study. Of note, the workflow from Santana *et al.* focused on methods that take peak-level fold-changes across condition as input, and therefore did not consider methods that compute per-sample activity scores, such as chromVAR. This, along with differences in the methods included and datasets used (in particular, a large number of H3K27ac datasets versus ATAC-seq simulations in our case), make the two studies complementary, although largely agreeing on some common observations.

### 4.2 Open questions

An issue that was barely touched in our study is the definition of the consensus peaks on which to work. How peak calling should best be done in ATAC-seq, especially across samples, is still an open debate. Our attempt to restrict to peaks called on nucleosome-free fragments, in particular the fact that counting all or only nucleosome-free fragments on nucleosome-free peaks both give similar results, suggests that the peaks, rather than the counting, is the decisive factor. However, further study will be needed to establish what type of peak calls, and whether leniency or stringency in the thresholding, is best for TF activity inference.

We have shown that using motif archetypes rather than an extensive set of similar motif does not generally improve results except for the multivariate approach, but the impact of the motif collection (or binding sites or other proxy to it) used will require further research. Our simulations indicate that, as expected, with constant perturbation strength the difficulty in detecting a difference in TF activity is strongly related to how good a proxy for binding the motif is. Unfortunately, it is well known that motifs are in general poorly informative of TF binding. Further work should be put in comparing motif collections against other forms of binding site predictions, or collections of experimental binding sites [46].

Finally, our benchmark is based on datasets with a small number of samples (2-3 per experimental group). While such a design is very common, it is possible that some methods will benefit more from larger sample sizes. In particular, it is plausible that the sample-permutation mode of diffTF, and more generally methods that compute per-sample scores, would benefit more from increased sample sizes.

## Supporting information

**S1 Appendix. Supplementary Figures A to S.**
(PDF)

**S1 Table. Name and description of each method variation tested.**
(XLSX)

## Acknowledgments

We thank Lukas Burger for providing the BANP motif.

## Author Contributions

**Conceptualization:** Pierre-Luc Germain.

**Data curation:** Felix Ezequiel Gerbaldo.

**Formal analysis:** Felix Ezequiel Gerbaldo, Emanuel Sonder, Pierre-Luc Germain.

**Investigation:** Felix Ezequiel Gerbaldo, Emanuel Sonder, Vincent Fischer, Selina Frei, Jiayi Wang, Pierre-Luc Germain.

**Methodology:** Emanuel Sonder, Pierre-Luc Germain.

**Project administration:** Pierre-Luc Germain.

**Resources:** Vincent Fischer, Selina Frei, Katharina Gapp.

**Supervision:** Mark D. Robinson, Pierre-Luc Germain.

**Writing – original draft:** Felix Ezequiel Gerbaldo, Emanuel Sonder, Pierre-Luc Germain.

**Writing – review & editing:** Selina Frei, Katharina Gapp, Mark D. Robinson, Pierre-Luc Germain.

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
