## [Decision Letter · Decision Letter 0]

3 Jun 2024

Dear Dr Germain,

Thank you very much for submitting your manuscript "On the identification of differentially-active transcription factors from ATAC-seq data" for consideration at PLOS Computational Biology.

As with all papers reviewed by the journal, your manuscript was reviewed by members of the editorial board and by several independent reviewers. As you will see, the reviews are encouraging, but the reviewers raised several important issues. In light of the reviews (below this email), we would like to invite the resubmission of a significantly-revised version that takes into account the reviewers' comments.

We cannot make any decision about publication until we have seen the revised manuscript and your response to the reviewers' comments. Your revised manuscript is also likely to be sent to reviewers for further evaluation.

Sincerely,

Shaun Mahony

Academic Editor

PLOS Computational Biology

Sushmita Roy

Section Editor

PLOS Computational Biology

Reviewer's Responses to Questions

**Comments to the Authors:**

Reviewer #1: The study of Gerbaldo et al. describes a thorough benchmarking study comparing methods for inferring the activity of transcription factors from differential chromatin accessibility between conditions, such as protein knockout or overexpression.

The overall design and structure of the study are very careful, and the execution of the analysis seems very convincing, from exhaustive supplementary figures to the organization of the underlying data and code. I strongly support the publication of the manuscript, as, despite the narrow topic and heavily technical nature of the study, this is a great practical example of performing a comprehensive benchmarking study, which includes simulations, previously published data, and a newly generated experimental dataset.

I consider the manuscript to be mostly publication-ready, although introducing minor edits could be useful here and there. Please find more details below. Of note, I am not a native English speaker thus some of my suggestions might be non-relevant.

Sincerely yours,

Ivan V. Kulakovskiy

General

Throughout the text, you use the term 'motif' ambiguously: (1) to denote the model of the DNA pattern describing the TF-DNA binding specificity and (2) to denote particular pattern occurrences in the sequences of ATAC-Seq peaks. I strongly suggest differentiating the respective objects explicitly: please use the term 'motif' or 'motif model' to describe the pattern (e.g. the position weight matrix) and use 'motif hits' or 'motif occurrences' or 'motif matches' when talking about particular 'words' yielded by sequence scanning (motif finding, pattern matching). This ambiuity is a general issue in the literature and would be great to avoid. Particularly, this makes it difficult to properly understand the description of the 'network score' (line 237).

Another general problem comes with 'data not shown' statements, e.g., it is unclear what you mean by 'the best results' (line 127) or what are the 'other technical sources of variation' (lines 413-414). Please consider briefly describing the underlying data e.g. with a few examples.

Suggested additions

While the technical results are described in a very detailed way, the Discussion could include a special section dedicated to interpreting the observed effects. First, regarding the NFKB1 TRAFTAC data: should we treat the low-to-inconsistent effect for the NFKB1 motif as a typical outcome given what is known on its cellular role? Finding the members of the AP-1 complex among the most significant TFs linked with the differential chromatin accessibility is not surprising, and a brief comparison to the other benchmarked datasets could be informative, i.e., whether those motifs are commonly detected on top of the list. Exploring the composition of the typical 'false positives' could be also informative in this context.

Similarly, it seems interesting to hear your interpretation of striking differences between close family members, such as GATA1-GATA2 and RUNX1-RUNX2. Is it a direct consequence of a limited overall magnitude of change in chromatin accessibility for one of two similar TFs? Could it be linked to a difference in protein expression? This might be helpful to highlight and explain the 'difficult' datasets, especially for pairs of TFs sharing similar binding specificities and, thus, similarly affected by any issues related to motif scanning.

Motif scanning

Using FIMO with default settings is appropriate as an off-the-shelf approach, but exploring if using strict or weaker thresholds would affect the 1-2 top-scoring methods might be informative, e.g., using NFKB TRAFTAC data as a brief case study. This might be particularly interesting given monaLisa's solid performance and internal motif-finding procedure.

Minor comments regarding the text

The following word choices seemed strange to me and I suggest rechecking and, if necessary, rephrasing the respective statements and/or providing more details:

'analytic variations' // Author summary

'the impacts of a novel method' // Author summary

'regulate a given transcriptional signature' // line 4

'less biased' // line 21

'peak samples' // line 68

'each rule' // lines 146-147

'we had some difficulties running the code' // lines 197-198, please provide some details

'Benjamini, Hochberg, and Yekutieli' // lines 247-248, do you mean "BH" (aka "FDR" ) or "BY"?

Minor comments regarding Figures and Figure labels

Figure 1: Please provide an extra (supplementary?) table summarizing all variants of methods assessed in the benchmark (rows in panel A). The current naming scheme is not uniform and hard to track throughout the manuscript. FDR on panel B (Y-axis) can be also mistaken for adjusted P-values. Please also spell out the plot labels, i.e. "Adj. P-value" instead of "p.adjust".

Figure 2: please rephrase 'true motif ranks' to reduce ambiguity. It can be read as 'ranks of true motifs' or 'true ranks of various motifs', which have dramatically different meanings in the context of your benchmarking setup.

Figure 5: 'simes' should be probably capitalized.

Reviewer #2: The manuscript titled "On the Identification of Differentially Active Transcription Factors from ATAC-seq Data" delves into the performance evaluation of TF enrichment analysis methods. The authors aimed to determine the optimal pipeline for TF enrichment by benchmarking background selection, variability normalization, incorporation of factors in multivariate linear models for differential analysis, and the combination of multiple tools to adjust p-values. It's noteworthy that quantile normalization yielded the best results for z-score normalization.

TF analysis presents a challenging question to address within a single manuscript, and it's not surprising that this study leaves several questions unanswered. Key steps in TF enrichment analysis include motif selection, definition of binding sites, fragment count, background correction, counts normalization, model selection, and enrichment analysis. While the authors covered most of these steps, there's a lack of thorough discussion on motifs. This oversight led to some expected results, as noted in lines 587-595. The authors should explain why they chose HOCOMOCO over other databases and address how motifs generated from different methods could introduce bias in TF binding site prediction. Additionally, merging similar motifs is a crucial step in motif cleaning that should not be overlooked.

It's important to recognize that there is no ground truth for TF enrichment analysis. The authors attempted to create a method for generating "true TFs," which could be a significant advancement in the field. Further discussion on "true TFs" in the manuscript, including aspects such as the distance distribution relative to transcription start sites and co-localizations among the binding sites of "true TFs," would greatly enhance the depth of the analysis.

Would it be beneficial to consolidate the software used in the comparison into a summarized table for clarity?

Additionally, it's intriguing to note that all footprint-based methods did not yield results for the "true TF." I wonder if the footprint-based method could potentially compensate for biases in chromatin accessibility-based methods. Could it be feasible to include the footprint for the NFkB motif under both conditions in the section discussing TRAFTAC-mediated TF degradation?

Here are some minor points that need to be addressed:

1. Detailed parameters are missing in lines 62-72.

2. I couldn't find a description for reads shift in the method, although the author may have mentioned it.

3. If possible, could you add the TFEA (https://doi.org/10.1038/s42003-021-02153-7) package to the comparison?

4. Would it be better to describe fold-changes by formula in line 268?

5. Many figure legends are missing, such as Figure S2D and Figure S4.

6. How did the authors conclude that the two metrics show strong agreement in Figure S2D?

7. There are some typos, such as "FigS2B" in line 371 and "FigS3C-E" in line 426.

8. There seems to be a figure repeated in Fig4AB and FigS7B.

9. It's challenging to see the red circles for NFKB1 and NFKB2 in Fig5A.

Reviewer #3: Review

The authors present a very nice benchmarking study for testing computational tools for identifying TF motif activity based on ATAC-seq data. They carefully selected a set of studies that specifically perturb only one TF ano d then measured ATAC-seq data. To further validate their findings, they simulated the removal of a TF by using ChIP-seq data of the TF to predict where it binds, and then specifically down sampled the ATACseq signal at these loci. They have done a great job in assembling a very thoughtful ground truth, both simulated and real data.

Major comments:

- while the authors tested the whole range of methods on the real data, they only tested a couple of methods on their simulated data. The simulated data, however is much better controlled in terms of what effects are expected, so it would be important to show all methods on the simulated data. One caveat of the real data - especially the K/O datasets that had the TF down for 72h is that there may be lots of indirect effects as well so testing the methods on the simulated data would be more fair to compare all method, because there, the authors know exactly what to expect.

- The authors mention they ranked the TFs by fold-change, did they consider absolute fold-change? This is particularly important for TFs that may act as repressor in the real datasets (not in the simulated) because one would expect a loss of accessibility for repressors (as shown e.g. in the diffTF paper). Maybe the reason Runx2 is not detected by any method is because it is a repressor and accessibility is lost upon its deletion?

- diffTF should not be used in permutation mode for these small sample sizes, this is completely meaningless. Also it is meaningless to increase permutations to 1000, that just means many permutations will be exactly the same

- The authors should emphasise better that ChromVar only performs well when combined with their additional extension using the limma model, it should be very clear that ChromVar on its own (I’m assuming that’s chromVAR:differentialDeviations) performs quite badly. This is important for readers to understand, not that they use chromVar in default.

- Many TFs have similar motifs, the authors should consider this, especially for the precision measurement in Fig 1B, but maybe also for the rank based analysis. I could envision a similar analysis as the network based analysis (which is very cool by the way!) could be done for TFs with similar motifs

Minor comments:

- the references are not sorted according to appearance

- the ordering in Figure 1A is not very clear, how were the methods ordered? It seems some of the well-performing methods are lower than some of the more poorly performing methods. If it is based on the sum of the ranks, maybe the Runx2 should be taken out to avoid skewing based on arbitrarily high ranks?

- I’m wondering whether it is necessary to have all the variations of one method in the main figure, or whether it would be sufficient to show the default and the best-performing addition to one method in the main figure and the rest in the supplement. It would make the main figure 1 easier to read - but this is of course a decision the authors should take.

**Have the authors made all data and (if applicable) computational code underlying the findings in their manuscript fully available?**

Reviewer #1: Yes

Reviewer #2: Yes

Reviewer #3: Yes

PLOS authors have the option to publish the peer review history of their article (what does this mean?). If published, this will include your full peer review and any attached files.

Reviewer #1: **Yes: **IVAN V KULAKOVSKIY

Reviewer #2: No

Reviewer #3: **Yes: **Judith Zaugg
---

## [Decision Letter · Decision Letter 1]

23 Sep 2024

Dear Dr Germain,

We are pleased to inform you that your manuscript 'On the identification of differentially-active transcription factors from ATAC-seq data' has been provisionally accepted for publication in PLOS Computational Biology.

Best regards,

Shaun Mahony

Academic Editor

PLOS Computational Biology

Sushmita Roy

Section Editor

PLOS Computational Biology

Reviewer's Responses to Questions

**Comments to the Authors:**

Reviewer #1: The authors carefully and thoroughly addressed the issues of the original manuscript and I consider the current version fully suitable for publication.

Reviewer #2: The authors have done a very nice job in addressing this reviewers comments.

Reviewer #3: The authors have addressed my comments very nicely. In the meantime we have replied to the error message posted for diffTF on the simulated data.

Well done,

Judith Zaugg

**Have the authors made all data and (if applicable) computational code underlying the findings in their manuscript fully available?**

Reviewer #1: Yes

Reviewer #2: Yes

Reviewer #3: Yes

PLOS authors have the option to publish the peer review history of their article (what does this mean?). If published, this will include your full peer review and any attached files.

Reviewer #1: **Yes: **IVAN V KULAKOVSKIY

Reviewer #2: No

Reviewer #3: **Yes: **Judith Zaugg

---

## [Editor Report · Acceptance letter]

9 Oct 2024

PCOMPBIOL-D-24-00394R1 

On the identification of differentially-active transcription factors from ATAC-seq data

Dear Dr Germain,

I am pleased to inform you that your manuscript has been formally accepted for publication in PLOS Computational Biology. Your manuscript is now with our production department and you will be notified of the publication date in due course.

With kind regards,

Dorothy Lannert
